# Sustainable Knowledge Investment Increases Employment and GDP in the Spanish Agricultural Sector More Than Other Investments

**Carolina Cosculluela-Martínez** 

Applied Economics I Department, Law and Social Sciences Faculty, University Rey Juan Carlos, 28032 Madrid, Spain; carolina.cosculluela@urjc.es; Tel.: +34-620-630-299

**Abstract:** Investment in every type of asset increases GDP and net employment differently. This paper compares the effect produced by a permanent unitary shock in Sustainable Knowledge for the Primary Sector (SKPS) on the Spanish employment and GDP growth with the effect produced by the other fourteen capital stock types. The methodology used is a Vector Error Correction Model (VECM), where the complementary capital can affect SKPS instantaneously. The results suggest that SKPS produces the second-highest, short and long-term effects on both labor and production, per Euro invested; moreover, the investment of 4.3 thousand euros is retrieved in the first year and increases net employment in one person after four years. Accordingly, the 5 million Euro Budget to invest in sustainable machinery and processing techniques increases net employment by 827 employees.

**Keywords:** sustainable knowledge; sustainable development; productivity; Gross Domestic Product; social labor; macroeconomic tool; investment policies

## 1. Introduction

Aschauer's research [1,2] focused on quantifying the effects of capital investment on the economy. In recent years there has been a significant number of papers on the critical topic of estimating the productivity of the agricultural sector [3–7]. In the meantime, in Spain, the percentage of GDP invested in machinery for the primary sector has decreased from 46.22% (1982–1986) to 17.57% (2002–2005). However, the change has come, the Spanish Government has increased the quantity for the sector by 1.3%, and there are 5 million euros for investment in sustainable machinery and processing techniques [8]. Spain is an agricultural country, the second in Europe. The agricultural, fishing, and farmer sector produces more than 10%, directly or indirectly, of the GDP [9]. The agricultural sector is one of the economic engines affecting exports and employment.

The first goal of this paper is to calculate the effects of a structural unitary shock in sustainable knowledge for the primary sector capital stock (SKPS) on the growth of GDP and net employment. Is knowledge, advances in technology towards the Agenda 2030, productive for the sector? The second objective is to compare these effects with those of other types of capital stock. The aim is to provide answers to the following questions: Is investment in SKPS productive in Spain, or does the primary sector show signs of having technological obsolescence? If so, are EU funds for investment sufficiently productive in Spain to contribute to the so-called European Cohesion?

Until the year 2000, when Pereira [10] itemized five different types of public investment, not considering all existing capitals, there had been no research on the effects of investment on different types of public capital.

This paper compares the effects of SKPS capital stock with 14 types of capital stocks, considering former capitals. The other capitals analyzed are (1) housing capital stock, (2) aggregated construction,

(2.1) road infrastructures, (2.2) public water infrastructures, (2.3) railways, (2.4) airports, (2.5) port infrastructures, (2.6) civil infrastructures, (2.7) other constructions, (3) aggregated transport, (3.1) motor vehicles, (3.2) other vehicles, (4) aggregated machinery capital stock, and (4.2) metal machinery capital stock.

Determining whether investing is productive in terms of GDP and employment has been a matter of academic interest for some time (Tables 1–3 provide the results obtained in previous studies and the types of data and methodologies used). The contribution of this paper to the field is the use of a particular Vector Error Correction Model in a meticulous disaggregated analysis.

**Table 1.** Results compared with others obtained in the previous empirical literature.

| Methodology | Author | Year | Elasticity | | Scope | Data | |
|---|---|---|---|---|---|---|---|
| | | | $\alpha_Y$ | $\alpha_L$ | | Series | Type |
| U-f(Y) | (Aschauer) [1,2] | 1989 | 0.39 | | National USA | 1949–1985 | (St)(P)(I: transport) |
| U-f(Y) | (Munnell) [11] | 1990 | 0.34 | | National USA | 1948–1987 | (St)(PI)(I: transport) |
| U-f(Y) | (Munnell, Cook) [12] | 1990 | 0.15 | | States USA | 1970–1986 | (St)(PI)(I: transport) |
| U-f(Y) | (Tatom) [13] | 1991 | 0.146 | | National USA | 1974–1987 | (St)(P)(I: transport and prices of energy) |
| U-f(Y) | (Eberts) [14] | 1997 | 0.15 | | States USA | 1988–1992 | (St)(P)(I: transport) |
| VAR | (Pereira, Roca-Sagalés) [15] | 2003 | 0.523 | | National SP | 1970–1995 | (St)(P)(I: transport: roads and highways, ports, airports and railways, communications) |
| VAR | (Pereira, Flores de Frutos) [16] | 1999 | 0.63 | 0.04 | National USA | 1956–1989 | (St)(P)(I: core infrastructure, buildings and equipment, road infrastructure, transport, airport infrastructure, gas and electricity, water and sewage systems, educational buildings, police, justice, administration, etc.) |
| VAR | (Pereira) [10] | 2000 | 0.005 | 0.004 | National USA | 1956–1997 | (PI)(I: conservation structures, development structures, and civilian equipment) |
| VAR | (Pereira) [10] | 2000 | 0.02 | 0.003 | National USA | 1956–1997 | (PI)(I: education buildings, hospital buildings, and other buildings (industrial, general office, police and fire stations, etc.) |
| VAR | (Pereira) [10] | 2000 | 0.009 | −0.01 | National USA | 1956–1997 | (PI)(I: sewage and water supply systems) |
| VAR | (Pereira) [10] | 2000 | 0.02 | 0.01 | National USA | 1956–1997 | (PI)(I: electric and gas facilities, transit systems, airfields, etc.) |
| VAR | (Pereira) [10] | 2000 | 0.006 | −0.006 | National USA | 1956–1997 | (PI)(I: highways and streets) |
| VAR | (Pereira) [10] | 2000 | 0.04 | 0.007 | National USA | 1956–1997 | (PI)(I: core infrastructure) |
| VAR/EC | Pereira [17] | 2001 | 0.26 | 0.13 | OECD (12 countries) National USA | Differs from each country 1960s to 1980s | (PI)(I: public capital) |
| VAR/EC | Pereira [17] | 2001 | 0.04 | 0.035 | OECD (12 countries) National SP | Differs from each country 1960s to 1980s | (St)(PI)(I: core infrastructures, residential and non-residential buildings, road and highway infrastructure, transport, airport infrastructure, gas and electricity infrastructure, sewage and water supply systems, buildings for the police, justice, administration, education, etc.) (6) |
| VECM | (Flores de Frutos, Gracia-Díez et al.) [18] | 1998 | 0.43 | 0.02 | National SP | 1964–1992 | (St)(P)(I: transport and communications) |
| VECM | Cosculluela | | 1.08 | 2.20 | National SP | 1977–2005 | (St)(I: public capital) (1) |
| VECM | Cosculluela | | 0.91 | 1.88 | National SP | 1977–2005 | (St)(I: transport and communications) (2) |
| VECM | Cosculluela | | 0.74 | 1.50 | National SP | 1977–2005 | (St)(I: transport) (3) |
| VECM | Cosculluela | | 0.04 | 0.08 | National SP | 1977–2005 | (St)(I: road infrastructure) |
| VECM | Cosculluela | | 0.20 | 0.42 | National SP | 1977–2005 | (St)(I: other non-housing constructions) (4) |
| VECM | Cosculluela | | 0.12 | 0.25 | National SP | 1977–2005 | (St)(I: sewage and water supply systems) |
| VECM | Cosculluela | | 0.55 | 1.11 | National SP | 1977–2005 | (St)(I: Transit and airfields) (5) |

**Table 1.** *Cont.*

| Methodology | Author | Year | Elasticity | Scope | Data |
|---|---|---|---|---|---|
| | | | Source: Adapted from Munnell (1990) [11] | | |

Notes:

(1)  includes road infrastructure, public water infrastructures, railway infrastructures, airport infrastructures, port infrastructures, civil infrastructures, other transport material (goods and passengers) and education buildings, warehouses, hospitals, churches and other ecclesiastical buildings, etc.

(2)  includes road infrastructure, railway infrastructures, airport infrastructure, port infrastructure, other transport material (goods and passengers), and non-specialized machinery (hardware and software).

(3)  includes road infrastructure, railway infrastructures, airport infrastructure, port infrastructure, other transport material (goods and passengers).

(4)  includes education buildings, warehouse, hospitals, churches and other ecclesiastical buildings, etc.

(5)  includes road infrastructure, airport infrastructure, other transport material (goods and passengers).

(6)  As they used Argimón et al. computed capital stock data in first differences of log levels, they assumed data correspond to real net investment.

Keys: (U) static relations; (PI) public investment; (f(Y)) Production function; ($\alpha_Y$) Output elasticity to capital; ($\alpha_L$) Labor elasticity to capital; (I: transport) core investment infrastructure in transportation systems; (USA) United States; (SP) Spain; (St) capital stock; (P) Public; (VAR) Autoregressive Vector; (EC/VEC) Vector Error Correction Model.

- The estimated elasticity of output to other constructions and transportation in Spain (1.08) is higher than its USA counterpart (0.63) estimated by Pereira and Flores [16] (0.63) using capital stock series and by Pereira [10,15] 0.04 and 0.26, respectively, using investment series.
- The estimated elasticity of output to capital stock of road, port, air, railway and communication networks (0.91) is higher than that obtained by Pereira and Roca-Sagales [15] (0.52), and the one estimated by Flores et al. [4] (0.43) in Spain. Pereira and Roca-Sagales [15] omit possible presence of co-integration relations between the variables without testing it.
- The estimated elasticity of output to capital stock of road, port, air and railway networks (0.74) is higher in Spain than that obtained in different countries and regions around the world. The highest estimated elasticity of those countries and regions found in the literature is (0.71).
- The estimated elasticity of output to road network in Spain (0.04) is higher than the one obtained by Pereira [10] (0.006) using investment series of USA.
- The estimated elasticity of output to non-residential constructions in Spain (0.2) is higher than the one obtained by Pereira (2000) [10] (0.02) using investment series of USA.
- The estimated elasticity of output to sewage and water supply systems in Spain (0.12) is higher than the one obtained by Pereira (2000) [10] (0.009) using investment series of USA.
- The estimated elasticity of output to transit systems and airfields in Spain (0.51) is higher than the one obtained by Pereira (2000) [10] (0.02) using investment series of USA, including gas, electricity, and transit systems and airfields investment.

**Table 2.** Marginal productivity and return rates in Spain and the USA.

| Author | Year | Investment | Marginal Productivity | Return Rate |
|---|---|---|---|---|
| Pereira | 2000 | Public capital | $4.46 | 7.8% |
| | | Core infrastructures (highways and streets) | $1.97 | 3.4% |
| | | Core infrastructures (electric and gas facilities, transit systems, airfields . . . ) | $19.79 | 16.1% |
| | | Core infrastructures (sewage and water supply systems) | $6.35 | 9.7% |
| | | Education buildings, hospital buildings, and other buildings (industrial, general office, police and fire stations, etc.) | $5.53 | 8.9% |
| | | Conservation structures, development structures, and civilian equipment | $4.06 | 7.2% |
| Pereira and Flores | 1999 | Core infrastructure, buildings and equipment, transportation vehicles, road and airport networks, gas electricity, sewage and water supply systems, education buildings, hospital buildings, and other buildings (industrial, general office, justice, administration, police and fire stations, etc.). | $0.65 | |
| | | SPAIN | | |
| Pereira and Roca-Sagales | 2003 | Transportation (roads, ports, airports, and railroads) and communications owned by national, regional administrations. | 2.892€ | 5.5% |
| Cosculluela | 2009 | Core infrastructures, buildings and equipment, transportation vehicles, road and airport networks, gas electricity, sewage and water supply systems, education buildings, hospital buildings, and other buildings (industrial, general office, justice, administration, police and fire stations, etc.). (1) | 15.34€ | 15.15% |
| | | Public capital (1) | 15.34€ | 15.15% |
| | | Core infrastructures (highways and streets) | 1.34€ | 15.82% |
| | | Core infrastructures (transit systems, airfields) (2) | 30.28€ | 14.81% |
| | | Core infrastructures (sewage and water supply systems) | 13.54€ | 16.63% |
| | | Education buildings, hospital buildings, and other buildings (industrial, general office, police and fire stations, etc.) | 1.59€ | 16.21% |
| | | Road transportation, ports, airfields and railways and communications (3) | 15.93€ | 14.93% |

**Table 2.** *Cont.*

| Author | Year | Investment | Marginal Productivity | Return Rate |
|--------|------|-----------|----------------------|-------------|

Marginal productivity is computed following Pereira [10] as the total added product obtained by the investment of one monetary unit along the ten first periods of time, in order to compare the results. That is, the total added production of the first 10 periods (caused by an increase in one transitory percentage point in the rate of growth) divided by the effect that the increase of one percentage point has produced in the capital series. The rate of return associated to this marginal productivity is an indicator of profitability, considering the effects in previous periods. It has been calculated as the rate of return that converges to that marginal productivity.

(1) Includes road infrastructures, public water infrastructures, railway infrastructures, airport infrastructures, port infrastructures, civil infrastructures, other transport material (goods and passengers) and education buildings, warehouse, hospitals, churches and other ecclesiastical buildings, etc.

(2) Includes road infrastructures, airport infrastructures, other transport material (goods and passengers).

(3) Includes road infrastructures, railway infrastructures, airport infrastructures, port infrastructures, other transport material (goods and passengers).

- Road networks and buildings for public services have the lowest marginal productivities in Spain and in the USA.
- Investment in public capital stock—non-residential buildings and transportation of people and goods—is much more productive in Spain than in the USA; during the 10 first periods, one euro in Spain produces €15,35 while one dollar produces $4.46, according to Pereira [10], and $0.65 according to Pereira and Flores (1999). Thus, the rate of return of public capital in Spain is 15.15% while in the USA it is 7.8%.
- On the one hand, transit capital and airfields are the investment with the highest marginal productivity in Spain (€30.28, base year 2000) and in the USA ($19.79), with corresponding rates of return of 14.81% and 16.1%, respectively. On the other hand, road transportation networks have the lowest marginal productivity, €1.34 (base year2000) and $1.97, respectively. Thus, the results in this paper for marginal Spanish productivity in road, port, air, train, and communication networks suggest €15.93 (base year 2000), while the result in Pereira and Roca-Sagales is €2.89 (base year 1986).
- Sewage and water supply systems have a marginal productivity of €13.54 (base year 2000), and its corresponding rate of return is 16.63% in Spain, while in the USA the marginal productivity is $6.35, and its corresponding rate of return is 9.7%.
- A one euro investment in construction, buildings for public services, in Spain produces, along 10 periods, €1.59 (base year 2000) of GDP, with an associated rate of return of 16.21%; on the other hand, in the USA, one dollar produces $5.53 of marginal production, with an associated rate of return of 8.9%.

**Table 3.** Long-term accumulated elasticity of private-sector variables with respect to public investment in the USA (Pereira, [10]).

| Investment | Rate of Return | Marginal Productivity | Elasticity | | |
|------------|---------------|----------------------|------------|--|--|
| | | | Output | Employment | Investment |
| Aggregated Public Investment | 7.8% | $4.46 | 0.04253 | 0.00735 | 0.22909 |
| | | | (0.025,0.045) | (−0.062,0.014) | (0.004,0.229) |
| Core infrastructures (streets and highways) | 3.4% | $1.97 | 0.0055 | −0.0057 | 0.01154 |
| | | | (0.002,0.006) | (−0.008,−0.004) | (−0.036,0.012) |
| Core infrastructures (electric and gas facilities, transit systems, airfields, etc.) | 16.1% | $19.79 | 0.02103 | 0.01143 | 0.09455 |
| | | | (−0.0008,0.021) | (−0.005,0.011) | (−0.047,0.104) |
| Core infrastructures (sewage and water supply systems) | 9.7% | $6.35 | 0.00856 | −0.01159 | 0.01239 |
| | | | (−0.0058,0.01) | (−0.012,−0.005) | (−0.058,0.017) |
| Education buildings, hospital buildings, and other buildings (industrial, general office, police and fire stations, etc.) | 8.9% | $5.53 | 0.01732 | 0.00285 | 0.02174 |
| | | | (0.0049,0.017) | (−0.008,0.0029) | (−0.12,0.025) |
| Conservation structures, development structures, and civilian equipment | 7.2% | $4.06 | 0.00491 | 0.00392 | 0.06874 |
| | | | (0.002,0.0056) | (0.0024,0.0056) | (0.021,0.069) |

Pereira computes the marginal productivity multiplying the output to public investment ratio for the last ten years by the elasticity of private output with respect to public investment. The rates of return that converge to that marginal productivity are calculated as the estimated profit of the investment taking into account the effects of those profits in the 20 previous periods. In parentheses are the estimated confidence bounds.

**Table 3.** *Cont.*

| Investment | Rate of Return | Marginal Productivity | Elasticity | | |
|---|---|---|---|---|---|
| | | | Output | Employment | Investment |

- The estimated elasticity of labor to non-residential constructions and transportation network in Spain (2.20) is higher than the one estimated by Pereira and Flores [16] (0.04) using capital stock series and by Pereira [10,15], 0.13 and 0.007, respectively, using investment series. It is also higher if it is compared to the one estimated by Pereira [10,15] (0.035) using capital stock series of the Spanish economy.
- The estimated elasticity of labor to transportation network (road, port, air, railway) and communications (1.88) in Spain is higher than the one obtained by Flores et al. [18] (0.02).
- The estimated elasticity of labor to road network in Spain (0.08) is higher than the one obtained by Pereira [10] (−0.006) using USA investment series.
- The estimated elasticity of labor to non-residential constructions in Spain (0.42) is higher than the one obtained by Pereira [10] (0.003) using USA investment series.
- The estimated elasticity of labor to sewage and water supply systems in Spain (0.25) is higher than the one obtained by Pereira [10] (−0.01) using USA investment series.
- The estimated elasticity of labor to transit systems and airfields in Spain (1.11) is higher than the one obtained by Pereira [10] (0.01) using investment series of USA, including gas, electricity, and transit systems and airfields investment.

This paper answers the following questions:

First, is an investment in SKPS productive compared with other capitals? Furthermore, how many periods are needed to retrieve the quantity invested?

Second, which types of investment have the highest short-, medium-, and long-term effects on employment? How much it is needed to increase net employment by one employee? Last, are there complementary effects found between employment and SKPS?

The analysis follows the methodology proposed by Cosculluela-Martínez and Flores de Frutos [19] to measure the effect of investing in housing in Spain and also partially used to find the weights in the Life Expectancy Index [20]. Other authors have either used this methodology or based analyses on its results [21–25].

The contribution of this paper lies in its comparison of the economic effects of different investment distributions, allowing the isolation of responses to shocks in the capital stock type studied and/or in the complementary capital stock. This isolation of responses makes it possible to study the effects, on labor and production, of different investment distributions at different periods. Therefore, the results obtained provide a useful macroeconomic policy instrument to quantify the effects of investment in SKPS capital stock compared with other capital stock types, information which may be helpful towards the goals of the Agenda 2030. The results provide enough evidence to determine that investment in SKPS is retrieved in the first period and increases net employment by one person after four years per investment of 4.3 thousand euros.

Section 2 of this paper (Materials and Methods) describes the version of the methodology used. Section 3 (Estimation of the Theoretical Model) presents the time series used, their statistical properties, and the empirical estimation of the theoretical model. Section 4 (Results) discusses the step response functions (SRFs) of output and employment. Section 5 (Discussion) presents a brief critical analysis of previous literature on the subject and provides a comparative analysis with the results obtained in previous papers. Section 6 concludes with a proposal on the results obtained and points out the limitations of the analysis.

## 2. Materials and Methods

Although the data of the Spanish economy available cover the timespan 1977- 2012, the data timespan was from 1977 to 2005. The worldwide crisis in 2008 changed the model adjustment completely, and there were not enough data from 2008 to 2012 to leave enough degrees of freedom to estimate two different models. This intervention analysis concluded not to modify the time series so the data would be accurate.

$Y_t$: Gross Domestic Product (GDP) obtained from the World Bank. Thousands of euros, the base year 2000.

$L_t$: Total employment (The employment of Ceuta and Melilla are not considered due to missing data in the first time periods.), measured in thousands of workers and obtained from the Spanish Labor Force Survey published by the National Statistics Institute (2006).

$K_{i_t}$: Capital stock data computed by the Instituto Valenciano de Investigaciones Económicas (IVIE) and published by the Banco Bilbao Vizcaya BBVA Foundation (Mas et al. 2007), where i = 1-4.3, according to BBVA-IVIE's second level classification. Thousands of euros, the base year 2000.

$\overline{K}_{i_t}$: Capital stock data computed by IVIE and published by the BBVA Foundation (Mas et al. 2007), excluding the isolated Capital Stock $K_{i_t}$. Thousands of euros, the base year 2000.

The main goal is to estimate the responses of $Y_t$, $L_t$, and $\overline{K}_{i_t}$ to an increase of $K_{i_t}$. Thus, the vector of variables which are used in this research is $W_t = \left(Y_t, L_t, K_{i_t}, \overline{K}_{i_t}\right)\prime$. As the variables of $W_t$ are stationary in the second differences of the variables I(2), lowercase variables $w_t = \left(y_t, l_t, \overline{k}_{i_t}, k_{i_t}\right)\prime$ represent the vector of first-differenced logged variables of $W_t$. The objective is to estimate the impulse response functions (IRFs) of $y_t$ and $l_t$ to a permanent unitary shock in $k_{i_t}$; the SRFs are computed by adding the IRFs.

In $w_t$ there are two types of variables: the vector of variables of fast reaction, vector $z_t = (y_t, l_t)'$ and vector $k_t = \left(k_{it}, \overline{k}_{it}\right)'$, which are the more rigid ones. The variables of $z_t$ react faster than those of $k_t$. It seems reasonable to think that a shock in $k_t$ (in period t) would have both instantaneous and lagged effects on the variables in $z_t$. However, a shock in period t in any variable of $z_t$ would only cause lagged responses in $k_t$ variables. This means that $k_t$ variables need time to react to changes in $y_t$ or $l_t$. Thus, $k_t$ levels are determined by past values of $z_t$, while $z_t$ values are determined by past and present values of $k_t$.

Formally, the behavior of vectors $z_t$ and $k_t$ can be represented as follows:

$$z_t = v_z(B)k_t + N_{z_t}$$
$$\pi_z(B)N_{z_t} = \alpha_{z_t} \tag{1}$$

$$k_t = v_k(B)z_t + N_{k_t}$$
$$\pi_k(B)N_{k_t} = \alpha_{k_t} \tag{2}$$

where $v_z(B)$ and $v_k(B)$ are $(2 \times 2)$ matrices of stable transfer functions:

$$v_z(B) = \begin{pmatrix} v_{y_{\overline{k}it}}(B) & v_{y_{kit}}(B) \\ v_{l_{\overline{k}it}}(B) & v_{l_{kit}}(B) \end{pmatrix} \text{ and } v_K(B) = \begin{pmatrix} v_{\overline{k}_{iy_t}}(B) & v_{\overline{k}_{il_t}}(B) \\ v_{k_{iy_t}}(B) & v_{k_{il_t}}(B) \end{pmatrix}$$

$\pi_z(B)$, $\pi_k(B)$ are polynomial matrixes in B, with the lag operator:
$\pi_z(B) = \pi_{o_z} - \pi_{1_z}B - \pi_{2_z}B^2$ whose elements are a $(2 \times 2)$ coefficient matrix.

Each function in $v_z(B)$ represents the response function of each variable $y_t$ and $l_t$ to shocks in $k_t$.

Together, $k_t$ variables have diverse yields. Complementary infrastructures $\overline{k}_{it}$ other than $k_{i_t}$ take longer to react than the infrastructure $k_{i_t}$ isolated.

It is worth noting that the empirical analysis shows significant contemporaneous correlations between some types of the capitals studied. The isolated infrastructure $k_{i_t}$ reacts immediately (in the same period) to variations in other infrastructures $\overline{k}_{it}$ and will continue reacting to those changes along several periods. However, other infrastructures $\overline{k}_{it}$ only show lagged effects to changes in the infrastructure studied and do not react in the same period. Thus, shocks in SKPS or in any other capitals produces delayed changes, from the second year onwards, in other infrastructures such as houses, roads, etc.

Mathematically this is expressed in Equation (3) as follows:

$$E(\alpha_{kt}\alpha_{kt}') = \Sigma_k = P_k\Sigma_k^*P_k' \tag{3}$$

where $P_K = \begin{pmatrix} 1 & 0 \\ -\beta & 1 \end{pmatrix}$ is the diagonalization matrix of $\Sigma_k$ and $\beta$ represents the slope in Equation (4).

$$\alpha_{kt} = \beta \alpha_{\bar{k}_t} + \alpha_{k_t}^* \tag{4}$$

Introducing this assumption, Equation (2) is represented as

$$P_k \Pi_k(B) k_t = P_k \Pi_k(B) \nu_k(B) z_t + \left[ P_k \alpha_{k_t} = \alpha_{k_t}^+ \right]$$

with

$$E\left( \alpha_{k_t}^+, \alpha_{k_t}^{+\prime} \right) = \Sigma_k^+ \tag{5}$$

as the diagonal.

Compacting Equations (1) and (5) results in

$$\begin{bmatrix} \Pi_z(B) & -\Pi_z(B)\nu_z(B) \\ -P_k \Pi_k(B)\nu_k(B) & P_k \Pi_k(B) \end{bmatrix} \times \begin{bmatrix} z_t \\ k_t \end{bmatrix} = \begin{pmatrix} \alpha_{z_t} \\ \alpha_{k_t}^+ \end{pmatrix} \tag{6}$$

with

$$E\left[ \begin{pmatrix} \alpha_{z_t} \\ \alpha_{k_t}^+ \end{pmatrix} \begin{pmatrix} \alpha_{z_t}' & \alpha_{k_t}^{+\prime} \end{pmatrix}' \right] = \begin{bmatrix} \Sigma_z & 0 \\ 0 & \Sigma_k^+ \end{bmatrix}$$

where:

- $\Sigma_z$ non-diagonal;
- $\Sigma_k^+$ diagonal;
- $E\left[ \begin{pmatrix} \alpha_{z_t} \\ \alpha_{k_t}^+ \end{pmatrix} \begin{pmatrix} \alpha_{z_t}' & \alpha_{k_t}^{+\prime} \end{pmatrix}' \right]$ block diagonal.

Model (6) is represented as

$$\Pi^+(B) w_t = \alpha_t^+ \tag{7}$$

with

$$E\left( \alpha_t^+ \alpha_t^{+\prime} \right) = \Sigma^+ \tag{8}$$

as the block diagonal.

Since $\Pi^+(0) = \begin{bmatrix} I & -\nu_{z0} \\ 0 & P_k \end{bmatrix} \neq I$, the stochastic Model (8) is not normalized in the sense of Jenkins and Alavi [26]. However, it can be normalized by pre-multiplying (8) by $\left[ \Pi^+(0) \right]^{-1}$:

$$\left( \Pi(B) = \left[ \Pi^+(0) \right]^{-1} \Pi^+(B) \right) w_t = \left( \left[ \Pi^+(0) \right]^{-1} \alpha_t^+ = a_t \right) \tag{9}$$

- $a_t$ is a $(4 \times 1)$ vector of structural shocks, which follows a white-noise vector process, with a diagonal contemporaneous covariance matrix $\Sigma$.

where (9) is the theoretical Model (10). The difference between them is that $\Sigma_z$ is not diagonal, which is the dependence of the variables in $\alpha_{z_t}$. Nevertheless, it permits estimating the response functions of each one of the elements of $z_t$ to a shock in $k_{i_t}$.

$$\Pi(B) w_t = a_t \tag{10}$$

Estimating (10) and its corresponding instant covariance matrix permits consistent estimation of the parameters in (7) and (8) (all mathematical details can be provided under request); i.e., $\left[ \Pi^+(B) \right]$

and $\Sigma^+$, and from them, the IRFs. Positions (1,4) and (2,4) of the polynomial elements in (12) will give the response functions of $y_t$ and $l_t$, respectively.

$$w_t = \Psi^+(B)\alpha_t^+ \tag{11}$$

with

$$\Psi^+(B) = \left[\Pi^+(B)\right]^{-1} = \Psi_0^+ + \Psi_1^+ B + \Psi_2^+ B^2 + \dots \tag{12}$$

In the following section, Equations (9) and (12) are estimated.

## 3. Estimation of the Theoretical Model

Before estimating the multivariate theoretical model, a univariate analysis is done. The software used to carry out the univariate analysis is called E-Views 9.

Univariate Analysis. All variables are I(2) according to the values of the Augmented Dickey-Fuller (ADF) test [27,28] (the Autoregressive Integrated Moving Average (ARIMA) univariate estimated models suggest that none of the series seems to be over differenced, and no MA terms are present) for a unit root in first and second differenced series. The analysis shows that no intervention analysis was needed (all ARIMA univariate analyses and ADF tests can be provided under request via email). No important outliers were present.

Co-integration. Methods from Johansen [29,30] were used to study whether there were co-integration relationships among the variables $\left(y_t,\ l_t,\ \bar{k}_{i_t}, k_{i_t}\right)$.

Results suggest that there was one co-integration equation $\xi_{1t}$, which involves GDP and employment growth rates isolating every type of capital stock. (Methods from Engle and Granger [31] were used to check Johansen results. All regressions, including all the variables and constant terms, have been estimated. The ADF test indicates that for every w vector isolating every type of capital stock, the residuals of the regression of $y_t$ on $l_t$, $k_{i_t}$, and $\bar{k}_{i_t}$ are I(0), according to Phillips and Ouliaris [32] critical values (95% critical value, −4.11). There are also I(0) when each capital series is not considered in the regression (95% critical value, −3.77) or when both of them are excluded (95% critical value, −3.37). Thus, when $l_t$ is not included in the regression of $y_t$ on each capital series, or on both of them, the ADF test indicates that residuals are I(1). The ADF test indicates that the residuals of the regressions of $l_t$, $k_{i_t}$, and $\bar{k}_{i_t}$ on the other variables in each set of variables that isolate every type of capital stock series are I(1), except when airport infrastructures is the isolated capital.)

$$\xi_{1t} = y_t - 0.47_{(0.05)}l_t - 0.02_{(0.001)}$$

Co-integration equation $\xi_{1t}$ can be interpreted as a stable, positive relationship between GDP and employment growth rates, where the disequilibrium in each period t is measured by $\xi_{1t}$.

If airport infrastructure capital stock $\left(k_{24_t}\right)$ is isolated, another co-integration equation $\xi_{2t}$ is found between variables of w vector. (When airport infrastructures is the isolated capital, the ADF test indicates that the residuals of the regression of $l_t$ on $y_t$, $k_{24_t}$, and $\bar{k}_{24_t}$ are I(0). In fact, in the regression of $l_t$ on $y_t$, $k_{24_t}$, and $\bar{k}_{24_t}$, the ADF value is −5.72, while in the regression of $l_t$ on $k_{24_t}$ and $\bar{k}_{24_t}$, of $l_t$ on $y_t$ and $\bar{k}_{24_t}$, of $l_t$ on $y_t$, and of $l_t$ on $\bar{k}_{24_t}$, the ADF values are respectively −2.30, −3.64, −3.10, and −2.17. Then, by isolating every type of capital, one co-integration relationship including $\nabla \ln Y_t$ and $\nabla \ln L_t$ is found, and when airport infrastructures is the isolated capital, another co-integration relationship that must include $\nabla \ln L_t$, $\nabla \ln Y_t$, and $\nabla \ln K_{24_t}$ is found.)

$$\xi_{2_t} = l_t - 1.39_{(0.15)}y_t - 0.27_{(0.06)}k_{24_t} + 0.04_{(0.01)}$$

where $\xi_{2_t}$ measures the volatility in the stable, positive relationship between employment, GDP, and airport infrastructure growth rates.

## 4. Results

*Estimation of the Multivariate Model*

The software used for the multivariate analysis is J-Multi 4.15.

Akaike information criterion (AIC) (AIC values can be provided under request via email) suggests that the variables follow a VAR(3). Therefore, VEC(2), on twice differenced variables, was estimated by Generalized Least-Squares (GLS). All non-significant parameters were set to be zero. AIC applied to the residuals of the model showed that $a_t$ followed a multivariate white noise process (all diagnoses of the estimated models can be provided under request via email). From $\widehat{\Sigma^+}$ the instant correlation matrix $\hat{\rho}$ is calculated, and $\hat{\Pi}^+(0)$ is estimated. $\hat{\Pi}^+(0)$ permits the estimation of (10) from (9). Model (10) will be calculated by pre-multiplying (9) by $\hat{\Pi}^+(0)$. Table 4 shows the resulting Model (10) adjusted to data. The compact model is presented in Table 4 as

$$\hat{\Pi}_w^+(B)w_t + c = \hat{a}_t^+$$

**Table 4.** Orthogonalized Reduced Forms.

| $\hat{\Pi}_w^+(B)w_t = \hat{a}_t^+$. | | |
|---|---|---|
| $\hat{\Pi}_w^+(B)$ | $w_t$ | $\hat{a}_t^+$ |
| $\begin{bmatrix} \hat{\Pi}_{w_{11}}^+(B) & \hat{\Pi}_{w_{12}}^+(B) \\ \hat{\Pi}_{w_{21}}^+(B) & \hat{\Pi}_{w_{22}}^+(B) \end{bmatrix}$ | $\begin{pmatrix} y_t \\ l_t \\ \bar{k}_{it} \\ k_{it} \end{pmatrix} + \begin{pmatrix} c_1 \\ c_2 \\ c_3 \\ c_4 \end{pmatrix}$ | $\begin{pmatrix} \hat{a}_{y_t}^+ \\ \hat{a}_{l_t}^+ \\ \hat{a}_{\bar{k}_it}^+ \\ \hat{a}_{k_it}^+ \end{pmatrix}$ |
| **Agricultural, Farm, and Fishing Machinery ($k_{4_1 t}$)** | | |
| $\hat{\Pi}_{w_{11}}^+(B)$ | | $\hat{\Pi}_{w_{12}}^+(B)$ |
| $\begin{bmatrix} 1 - 0.29B & -0.34B \\ 0 & 1 - 1.17B + 0.40B^2 - 0.23B^3 \end{bmatrix}$ | | $\begin{bmatrix} 6.04B - 2.07B^2 - 0.29B^3 - 3.68 & 0 \\ 9.12B - 3.12B^2 - 0.44B^3 - 5.56 & 0.06B - 0.06 \end{bmatrix}$ |
| $\hat{\Pi}_{w_{21}}^+(B)$ | | $\hat{\Pi}_{w_{22}}^+(B)$ |
| $\varnothing$ | | $\begin{bmatrix} 1 - 1.64B + 0.56B^2 + 0.08B^3 & 0 \\ 0 & B \end{bmatrix}$ |
| $\hat{\Pi}_{0,w}^+$ | $\hat{\rho}$ | $\hat{\Sigma}^+$ |
| $\begin{pmatrix} 1 & 0 & -3.68 & 0 \\ 1 & -5.56 & -0.06 \\ 1 & 0 \\ 0 & 1 \end{pmatrix}$ | $\begin{pmatrix} 1 & 0.81 & 0.80 & -0.07 \\ 1 & 0.85 & 0.15 \\ 1 & -0.05 \\ 1 \end{pmatrix}$ | $\begin{pmatrix} 1.30 \times 10^{-4} & 1.50 \times 10^{-4} & 2.29 \times 10^{-5} & -4.04 \times 10^{-5} \\ 2.62 \times 10^{-4} & 3.43 \times 10^{-5} & 1.30 \times 10^{-4} \\ 6.23 \times 10^{-6} & -6.56 \times 10^{-6} \\ 2.85 \times 10^{-3} \end{pmatrix}$ |

This is presented together with their corresponding $\widehat{\Sigma^+}$, $\hat{\rho}$, and $\hat{\Pi}^+(0)$ matrix, where c is a vector of constant terms.

The relations among all the variables are shown in Table 4. As explained in Section 2, IRFs can be found from the reduced form of Model (10) in Table 4. By adding up the IRFs, the SRFs are computed, as mentioned in the following section. The elasticity is calculated by dividing the SRFs of production, employment, or complementary capital by the SRFs of the capital of the shock.

Table 5 shows the calculated reactions of production, employment, complementary capital stock, the 70% confidence level Bootstrap bounds, and also the feedback effects, in percentage points, for each of the following 20 periods, to an increase in the level of the aggregated and disaggregated capital stock of one percentage point.

**Table 5.** Response functions (%) of each variable level to a shock in agricultural, farm, and fishing machinery capital stock.

| Period | LB | ln Y | UB | LB | ln L | UB | LB | ln $\bar{K}_{4-1}$ | UB | LB | ln $K_{4-1}$ | UB |
|---|---|---|---|---|---|---|---|---|---|---|---|---|
| 1 | 0.00 | 0.00 | 0.00 | 0.03 | 0.06 | 0.11 | 0.00 | 0.00 | 0.00 | 0.77 | 1.00 | 1.42 |
| 2 | 0.01 | 0.02 | 0.04 | 0.04 | 0.07 | 0.13 | 0.00 | 0.00 | 0.00 | 0.77 | 1.00 | 1.42 |
| 3 | 0.02 | 0.03 | 0.05 | 0.04 | 0.06 | 0.10 | 0.00 | 0.00 | 0.00 | 0.77 | 1.00 | 1.42 |
| 4 | 0.02 | 0.03 | 0.05 | 0.03 | 0.05 | 0.10 | 0.00 | 0.00 | 0.00 | 0.77 | 1.00 | 1.42 |
| 5 | 0.01 | 0.02 | 0.05 | 0.03 | 0.05 | 0.10 | 0.00 | 0.00 | 0.00 | 0.77 | 1.00 | 1.42 |
| 6 | 0.01 | 0.02 | 0.05 | 0.03 | 0.05 | 0.10 | 0.00 | 0.00 | 0.00 | 0.77 | 1.00 | 1.42 |
| 7 | 0.01 | 0.03 | 0.05 | 0.03 | 0.05 | 0.10 | 0.00 | 0.00 | 0.00 | 0.77 | 1.00 | 1.42 |
| 8 | 0.01 | 0.03 | 0.05 | 0.03 | 0.05 | 0.10 | 0.00 | 0.00 | 0.00 | 0.77 | 1.00 | 1.42 |
| 9 | 0.01 | 0.03 | 0.05 | 0.03 | 0.05 | 0.10 | 0.00 | 0.00 | 0.00 | 0.77 | 1.00 | 1.42 |
| 10 | 0.01 | 0.03 | 0.05 | 0.03 | 0.05 | 0.10 | 0.00 | 0.00 | 0.00 | 0.77 | 1.00 | 1.42 |
| 11 | 0.01 | 0.03 | 0.05 | 0.03 | 0.05 | 0.10 | 0.00 | 0.00 | 0.00 | 0.77 | 1.00 | 1.42 |
| 12 | 0.01 | 0.03 | 0.05 | 0.03 | 0.05 | 0.10 | 0.00 | 0.00 | 0.00 | 0.77 | 1.00 | 1.42 |
| 13 | 0.01 | 0.03 | 0.05 | 0.03 | 0.05 | 0.10 | 0.00 | 0.00 | 0.00 | 0.77 | 1.00 | 1.42 |
| 14 | 0.01 | 0.03 | 0.05 | 0.03 | 0.05 | 0.10 | 0.00 | 0.00 | 0.00 | 0.77 | 1.00 | 1.42 |
| 15 | 0.01 | 0.03 | 0.05 | 0.03 | 0.05 | 0.10 | 0.00 | 0.00 | 0.00 | 0.77 | 1.00 | 1.42 |
| 16 | 0.01 | 0.03 | 0.05 | 0.03 | 0.05 | 0.10 | 0.00 | 0.00 | 0.00 | 0.77 | 1.00 | 1.42 |
| 17 | 0.01 | 0.03 | 0.05 | 0.03 | 0.05 | 0.10 | 0.00 | 0.00 | 0.00 | 0.77 | 1.00 | 1.42 |
| 18 | 0.01 | 0.03 | 0.05 | 0.03 | 0.05 | 0.10 | 0.00 | 0.00 | 0.00 | 0.77 | 1.00 | 1.42 |
| 19 | 0.01 | 0.03 | 0.05 | 0.03 | 0.05 | 0.10 | 0.00 | 0.00 | 0.00 | 0.77 | 1.00 | 1.42 |
| 20 | 0.01 | 0.03 | 0.05 | 0.03 | 0.05 | 0.10 | 0.00 | 0.00 | 0.00 | 0.77 | 1.00 | 1.42 |

Notes: (*) Response functions of natural logarithms of each variable. LB and UB represent the lower and upper Bootstrap Bounds at a 70% confidence level, respectively.

The results can be summarized as follows:

1. Output responded with a lag of one year to a shock in SKPS, as well as housing capital stock, road, airport and railway networks, public water infrastructures, and civil infrastructures. A shock in other types of capital stock produced contemporaneous effects on output. These results were also obtained in previous literature [10]. The elasticity was not constant over time in most capitals, although the SKPS elasticity was (0.3) from the third period onwards.

2. Employment reacted quicker than output and did so instantaneously; it decreased its elasticity to SKPS over the time from 0.062 to 0.051, according to the natural decrement evolution of labor.

3. Results did not show feedback effects, with the exceptions of housing, other aggregated construction capital, and non-specific infrastructures. Thus, SKPS capital stock and the rest of the capital types studied remained at equilibrium just 1 percentage point over their initial level or less (airport infrastructures).

4. Effects on the complementary capital stock were not detected in SKPS only when the non-specific infrastructures and non-specialized machinery and software, and consequently also in aggregated machinery other than housing construction, received a shock. The weights on the total capital stock of the studied capital, in most cases, was too small to produce effects on its complementary capital.

## 5. Discussion

This section compares the results obtained with those in previous literature, whether estimating elasticity in static or in a dynamic framework.

From Aschauer's research in 1989 [1,2], until the beginning of the 21st century, there had been no research on the effects of investment on different types of public capital. Pereira [10] itemized five different types of public investment, but the investment time series applied did not allow considering all existing capitals, and he used investment series. It is worth noting that Pereira's breakdown of public capital into five categories, although excluding capital stock series, provided the most detailed results at that moment in time. Before Pereira [10], they divided the capital into only two types. He found that all variables were I(1) and, using Engle and Granger's method [31], tested the possible presence of co-integration equations, without finding any. The main conclusion was that

the slowdowns of the production in the United States during the 1970s was the reduction of public investment in infrastructures.

Table 1 shows the methodology, data, and the estimated elasticity to both production and employment obtained in the empirical literature on the subject. In Table 2, the first column shows the rate of return corresponding to the marginal productivity (column 2) of ten periods (long-term) to the monetary increase of output for a one-dollar investment in every type of public capital. The corresponding rate of return is the yield that converges to marginal productivity. The last three columns show the estimated elasticity, together with their corresponding confidence bounds. Table 4 also contains the marginal productivity and the return rates associated with the marginal productivities, i.e., the investment profit and the effect generated by the previous investment. Previous literature shows that elasticity of output to any investment estimated using USA data is lower than the elasticity of output using Spanish data, including the results obtained in this paper. Thus, according to the elasticity estimated, the investment profit of any asset is higher in Spain than in the USA. However, capital series and investment series are different in every country; therefore, some authors show the marginal productivity. Marginal productivity is typically computed as the total added product obtained from the investment of one monetary unit throughout the ten first periods. That is the total added production of the first ten periods, caused by a one transitory percentage point increase of the growth rate, divided by the effect that that increase has produced on the capital series. Therefore, this marginal productivity does not coincide with the marginal productivity of economic theory, so some authors also calculate the rate of return associated with it as an indicator of profitability. The "modified" rate of return becomes the estimated profit of the investment, taking into account the effects produced by the previous investment.

Table 2 shows a comparative analysis with the results obtained by and Pereira [10] Pereira and Flores de Frutos [16] in the USA, and by Pereira and Roca-Sagalés [15] in Spain. The highest marginal productivity is produced by an investment in the same type of capital in Spain using capital stock series than in the USA using investment series. Thus, the most and least productive capitals are the same in the USA and Spain. The rates of return show that investment in physical capital is more productive in Spain than in the USA.

Table 3 shows the estimated elasticity of labor to public capital in the range of 0.007–0.04. Thus, labor elasticity is lower than output elasticity. The results obtained in this paper suggest that the elasticity of labor is lower than the elasticity of output.

The estimated elasticity of labor to public capital (all types of capital included in the aggregated other constructions and transportation network) is 2.20. Thus, additional investment increases employment, so public capital and labor are complementary inputs in the production function.

From this vast amount of literature, we can conclude that the investment in SKPS has not been analyzed, and the Agenda 2030 places it as the second objective. As pointed out in the results section, the main conclusion is that, although during the first years other types of investment produce and increase net employment instantaneously, SKPS is the second most productive and employee-hiring investment from the second period onwards.

The literature does not estimate the effects of investment in SKPS compared with other capitals. More important is the number of periods needed to retrieve the quantity invested. Moreover, the types of investment that have the highest short-, medium-, and long-term effects on employment were not identified. When unemployment increases and public policies can turn around the curve, it is essential to know how much investment is needed to increase net employment in one employee in as many types of assets as possible to weigh the National Budget conveniently.

Comparing the capital stock types, the investment of 1 million euros of 2000 in SKPS increases net employment after 4 years by 229 people, nearly doubling the investment in railway infrastructures that leads to an increase by 128 people. Indeed, per 1 million euros in 2020 invested in sustainable machinery and processing techniques, net employment increases by 166 employees. While SKPS

produces 11.91€ per euro (base 2000) invested in 4 years and 24.73€ in 8 years, railway infrastructures only produce half as much, 6.28 and 12.15€, respectively, per euro (base 2000) invested.

The methodology used in this paper contributes to the field in that the two-step orthogonalization allows to affect the complimentary capital instantaneously, and that it considers the highest disaggregation of capital stock types for the Spanish economy to compare the effects of SKPS.

## 6. Conclusions

Applying the methodology proposed by Cosculluela-Martínez and Flores de Frutos [19] to data from the Spanish economy, it is possible to conclude that SKPS 1€ investment yields are €11.91 in the first four years (short-run), €24.73 after eight years (medium-run), and €47.32 after twenty years (long run).

Additionally, SKPS, together with airport infrastructures and other transport equipment, provides the highest employment (per euro invested) in every considered period.

SKPS, as well as investments in airport infrastructures, other transport equipment, railway infrastructures, and public water infrastructures, takes one period to be productive. Metal machinery, motor vehicles, or other constructions (non-specific), i.e., education buildings, warehouses, hospitals, churches, and other ecclesiastical buildings, produce instantaneous results.

Answering the second question, the investment required in SKPS to reduce the number of unemployed people by one person is almost 30 times less than the investment in non-specific constructions. Indeed, the investment needed to increase net employment by one employee after four periods varies from €3566.58 for investment in airfield infrastructures to €4372.77 for investment in SKPS or €6690.37 for other transport equipment, reaching €102,758.42 in non-specific construction infrastructures. To increase net employment by one employee after 20 periods, the investment needed in airport infrastructures is €3052, in SKPS €51,664.34, while in housing it is €93,690, in other aggregated constructions €63,800, and non-specific constructions €112.932,09, twice the investment needed in SKPS.

The answer to the third question—Do complementary effects exist between capital and employment when investing in SKPS?—is yes; investment in SKPS increases employment. Is there crowding in the effect of other types of capital? Externalities are present in the investment in other aggregated constructions and non-specific construction from the second period onwards and after one period in the investment in other machinery and software, but not in SKPS. These increase the effect on output and labor.

To summarize, SKPS is one of the two most productive and employee-hiring capitals of the 14 capitals analyzed in any timespan. It produces instantaneous effects on employment, although it needs one period to increase output. Thus, the results support the importance of the Spanish primary sector and the need for new technology to stimulate it. SKPS investment continues to be very productive, both in terms of output and in terms of employment. The sector does not show signs of saturation; in fact, it is the second most productive type of investment (per euro invested) after investment in airports. Therefore, according to the results of this paper, a 5 million euro investment after 4 years will increase net employment by 827 employees. Thus, it would be recommended to promote this type of investment.

Thus, the SKPS sector does not seem to be overinvested, and its productivity and capacity to employ people seem to be far from the inflection point. The results reveal that EU funds supporting the SKPS sector and encouraging European cohesion are productive and create employment, with 259 employees per million euros invested (the base year 2000). Results asses the second objective (2.a) of the Agenda 2030 to invest in technological development and agricultural infrastructure.

This paper contributes to the field in two ways; first the results of this paper could be expanded to countries with similar characteristics, and second, the methodology used to carry out the study can be used with data from any other country or region. The main limitation is that is not taking into account the effects of the period in which a global crisis can occur, so the results of the model can be

extended when the country is recovering from a crisis such as the 2008 economic crisis or the 2020 COVID-19 crisis.

**Funding:** No funding for the article.

**Acknowledgments:** I am grateful to anonymous referee and the editors for their useful comments. I am responsible for all the remaining errors.

**Conflicts of Interest:** The author declares no conflict of interest.

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
