# Peer review of "Sustainable Knowledge Investment Increases Employment and GDP in the Spanish Agricultural Sector More Than Other Investments"

_sustainability, doi:10.3390/su12083127_

Round 1

Reviewer 1 Report

The idea of the paper is interesting. The contents and results are comprehensive and I would encourage the authors to continue this important. I support their work and believe that it can be recommended for publication.

The methods used are adequate to the objective and allow to answer the main questions under study.

However, I believe that some aspects could be improved:

- I think the title of the manuscript requires upgrades as it is not correct in the present version;

- The abstract need to be polished significantly. It needs to be logical, and free of grammar errors. And don’t use abbreviations without giving them explanations like SKPS;

- For data, please explain why did you stop in 2005 ... we are in 2020;

- Please explain better the importance of the topic for the specific case of Spain;

- I would like to see some discussion about the potential generalization of the results obtained by the authors in this study;

- The last section should provide the study limitations and future research avenues.

The authors should re-examine their manuscript to improve it since the English needs to be improved in some places.

Author Response

“The idea of the paper is interesting. The contents and results are comprehensive and I would encourage the authors to continue this important. I support their work and believe that it can be recommended for publication.

The methods used are adequate to the objective and allow to answer the main questions under study.

However, I believe that some aspects could be improved:

  1. a) I think the title of the manuscript requires upgrades as it is not correct in the present version;
  2. b) The abstract need to be polished significantly. It needs to be logical, and free of grammar errors. And don’t use abbreviations without giving them explanations like SKPS;
  3. c) For data, please explain why did you stop in 2005 ... we are in 2020;
  4. d) Please explain better the importance of the topic for the specific case of Spain;

e1) I would like to see some discussion about the potential generalization of the results obtained by the authors in this study;

e2) The last section should provide the study limitations and future research avenues.

f) The authors should re-examine their manuscript to improve it since the English needs to be improved in some places.

I thank the reviewer for the valuable comments, helping me to improve the structure, content, and conclusions of the manuscript. 

  1. The title has changed by a title representing more the results of the analysis “Sustainable knowledge investment increases employment and GDP in the Spanish Agricultural Sector more than other investments.”
  2. I have polished the abstract, attending the clarity, correctness, grammar errors, and specifying the significance of the abbreviation SKPS in lines 9 to 18. 
  3. Line 109 explains the timespan selected: “Although the data of the Spanish economy available covers the timespan 1977- 2012, the data timespan is from 1977 to 2005. The worldwide crisis in 2008 changes the model adjustment completely, and there is not enough data from 2008 until 2012 to leave enough degrees of freedom to estimate two different models. Intervention analysis determines not to modify the time series to be fair to data.” The results of the analysis for the Spanish economy can be extended to similar countries for periods when no global crisis is taking place. Not altering the model by anomalous data is the main limitation of the study, noted in the last paragraph of the conclusions chapter. The point of taking into account only these periods is that the quantities, the effect of the investment, can be applied after an unstable period, such as the COVID-19 sanitarium crisis period, when the country is recovering from the crisis.
  4. I have explained the importance of the topic for the specific case of Spain in the first paragraph now in lines 23-31. 
  5. In the conclusions (last paragraph), I have explained the potential generalization of the results obtained in this study together with the limitations and future research avenues.

Reviewer 2 Report

Very nice revisions. This reviewer recommends acceptance.

Author Response

I thank the reviewer for Very nice revisions and the recommendation of acceptance. 

Reviewer 3 Report

Thank you for having the opportunity to read your paper. The theme of the paper is interesting. However, I am afraid the paper is very difficult to read for the reader. A brief look at the paper shows that the attachments are longer than the paper itself.
I believe that the paper has the following weaknesses:

a) Paper missing scientific background. It is not processed at all.
b) Chapter 2 of the Materials and Methods needs to be elaborated much more carefully. I am afraid that the author must clearly explain why he used the data from 1977-2005. Whether the results ending in the 2005 data series can be used at this time.
c) there is no justification as to why the author used the Dickey-Fuller test, why he did not use other tests.
d) formal formalities are not observed, the text is confusing
e) the discussion needs to be completely rewritten. In my opinion, it contains only a general statement, not a real scientific discussion
  I recommend to the author to consider how much material is to be published and how much the reader can adjust himself from the references. At this point, the paper is completely unsuitable for publication.

Author Response

I thank the reviewer for the valuable comments, helping me to improve the structure and content of the manuscript to make it easier to read and more precise in what It is essential to publish as attachments.
The whole paper has been rewritten to make it easier to read, and I left the critical attachments of the paper and positioning the tables in its corresponding place.
a) I rewrote the discussion chapter to provide an appropriate framework for hypotheses development, and I have integrated the hypothesis in the discussion section after their corresponding literature background, improving the narrative flow of logic of the discussion to show the novelty of the paper.
b) I have rewritten the chapter Material and Methods to explain more carefully the methodology. Line 109 explains the reason for taking into account data only from 1977 to 2005 and not including pre-crisis, or year of crisis: “Although the data of the Spanish economy available covers the timespan 1977- 2012, the data timespan is from 1977 to 2005. The worldwide crisis in 2008 changes the model adjustment completely, and there is not enough data form 2008 until 2012 to leave enough degrees of freedom to estimate two different models. Intervention analysis determines not to modify the time series to be fair to data.” The results of the analysis for the Spanish economy can be extended to similar countries for periods when no global crisis is taking place. Not altering the model by anomalous data is the main limitation of the study, noted in the last paragraph of the conclusions chapter. The point of taking into account only these periods is that the quantities, the effect of the investment, can be applied after an unstable period, such as the COVID-19 sanitarium crisis period, when the country is recovering from the crisis.
c) The Dickey-Fuller Test (ADF) is used to test the results of the Johansen co-integration test. Footnote number 6 clarifies now the results of applying the ADF test to the residuals of all the regressions, including and excluding variables. This methodology is much more robust than Johansen when the amount of data is small.
d) The text has been rewritten to make it easier to read.
e) I rewrote the discussion and removed some of the material from the paper. The purpose of leaving the tables comparing the previous literature is to point out the novelty of the paper, not only in the results but also in the way they are computed. The capital types were aggregated to compare the elasticity’ and the marginal productivity (together with the rates of return) with other authors. This comparison allows setting the contribution of the paper. The effects are higher in Spain than in the United States; those effects are different from the ones found in the literature. In both countries, the marginal productivity is similar for similar types of assets. However, the series is not calculated in the same way (ones consider the existing capital and others no) or for the same period.

With these changes, I hope you find it suitable for publication.

Reviewer 4 Report

Unfortunately the paper is of poor quality and very limited scientific value. In fact it is difficult what is the scientific problem that the Author is going to solve. Although there are two defined goals of the paper it is difficult to match the provided analysis, discussion and conclusions with these goals. Moreover the paper is very hard to read. In fact it is difficult to identify what is new in this paper. Although the paper is economic focused – the Author analyses very old data (years 1982-2005) what makes the results of these analysis very limited. Therefore – in my opinion – the paper shouldn’t be published in the Sustainability journal.

Author Response

I thank the reviewer for the valuable comments, helping me to improve the structure and content of the manuscript to make it easier to read and more precise in what It is essential and the contribution to the field of the paper.

I have rewritten the discussion chapter to provide an appropriate framework for hypotheses development, and I have integrated the hypothesis in the discussion section after their corresponding literature background, improving the narrative flow of logic of the discussion to show the novelty of the paper.

I have explained the importance of the topic for the specific case of Spain in the first paragraph now in lines 23-31 to assed its scientific value. In the conclusions (last paragraph), I have explained the potential generalization of the results obtained in this study together with the limitations and future research avenues. The purpose of leaving the tables comparing the previous literature is to point out the novelty of the paper, not only in the results but also in the way I estimate them. I aggregated the capital types to compare the elasticity’ and the marginal productivity (together with the rates of return) with other authors. This comparison allows setting the contribution of the paper. The effects are higher in Spain than in the United States; those effects are different from the ones found in the literature. In both countries, the marginal productivity is similar for similar types of assets. However, the series is not calculated in the same way (ones consider the existing capital and others no) or for the same period. For that reason, the appropriate methodology to calculate the effect of an investment in SKPS is a contribution to the scientific field.

Line 109 explains the reason for taking into account data only from 1977 to 2005 and not including pre-crisis, or year of crisis: “Although the data of the Spanish economy available covers the timespan 1977- 2012, the data timespan is from 1977 to 2005. The worldwide crisis in 2008 changes the model adjustment completely, and there is not enough data from 2008 until 2012 to leave enough degrees of freedom to estimate two different models. Intervention analysis determines not to modify the time series to be fair to data.” The results of the analysis for the Spanish economy can be extended to similar countries for periods when no global crisis is taking place. Not altering the model by anomalous data is the main limitation of the study, noted in the last paragraph of the conclusions chapter. The point of taking into account only these periods is that the quantities, the effect of the investment, can be applied after an unstable period, such as the 2020 COVID-19 sanitarium crisis period when the country is recovering from the crisis.

Thus, results apply to the next period when the unemployment is going to increase after the sanitarium crisis, and there is a need to face the Agenda 2030 goals.

With these changes, I hope you find it suitable for publication.

Reviewer 5 Report

The introduction section should be further described since the current introduction is more the description of a framework. 

The methods are analytical explained, though the programme use to carry out the data has not included. 

The discussion is more of a description what could be included in the results. Also, implications to the field, commenting the findings or giving suggestions for future work.

The conclusion have included the limitations of the research and acknowledgements of these. 

The references are wrongly placed, some should be separated via a comma. 

Example, line 24:[1], [2] should be [1,2]. 

Other thing that should be explained are pages 18 and 21, what is this? supplementary references?

Also, what theme font was used for the references? The line spacing seems odd, please consider reviewing the references. 

Author Response

I thank the reviewer for the valuable comments, helping me to improve the structure and content of the manuscript to make it easier to read and more precise in what It is essential and the contribution to the field of the paper.
I rewrote the discussion section to provide an appropriate framework for hypotheses development, and I have integrated the hypothesis in the discussion section after their corresponding literature background, improving the narrative flow. The introduction now shows the importance of the analysis of the Spanish economy.
In lines 519 and 615, I included the programs used for each part of the analysis.
The last paragraph of the conclusion section includes the contribution to the field, limitations, and future suggested work. I have explained the potential generalization of the results obtained in this study together with the future research avenues.
I checked all the references, format, duplicity, and the way I cite.

With these changes, I hope you find it suitable for publication.

Round 2

Reviewer 3 Report

Thank you for all the revisions. It can be seen that the author responded to all reservations (although the theoretical background is processed at the very limit of acceptability).

The paper is still very difficult for the reader to read, but this is because its format is not of sufficient quality. After formatting, the readability will be higher.

I believe that nothing needs to be corrected in the content of the paper.

Reviewer 4 Report

Thank you for your improvements and all explanations provided in the reply. I read your paper again. I think it can be published in current form.

Reviewer 5 Report

Following all the suggestions made, the manuscript has highly improved from the previous version making easier the reading and understanding of the findings.